# *Rothia nasimurium* as a Cause of Disease: First Isolation from Farmed Chickens

**DOI:** 10.3390/vetsci9120653

**Published:** 2022-11-22

**Authors:** Jiahao Zhang, Shaojiang Mo, Hu Li, Ruizhi Yang, Xiangjie Liu, Xiaoyue Xing, Yahui Hu, Lianrui Li

**Affiliations:** 1College of Animal Science and Technology, Tarim University, Alar 843300, China; 2Key Laboratory of Tarim Animal Husbandry Science and Technology, Xinjiang Production & Construction Corps, Alar 843300, China; 3Engineering Laboratory of Tarim Animal Diseases Diagnosis and Control, Xinjiang Production & Construction Corps, Alar 843300, China; 4College of Life Science and Technology, Tarim University, Alar 843300, China

**Keywords:** *Rothia nasimurium*, isolation and identification, multidrug resistance, chicken origin

## Abstract

**Simple Summary:**

*Rothia nasimurium* is an opportunistic pathogen. It can infect animals such as dogs, pigs, ducks, rabbits, and geese, and antibiotic susceptibility tests have confirmed that this bacterium has a multidrug-resistant phenotype. In January 2022, chickens at a poultry farm in China’s Xinjiang Uygur Autonomous Region became ill and died. In order to determine the cause of the disease in these poultry farm chickens, the isolation and identification of the pathogens in the livers and other internal organs of the sick chickens were performed. A bacterial strain was isolated from the livers of the diseased chickens. The isolated strain was identified to be *Rothia nasimurium*. The isolate was resistant to 17 antibiotics, including ciprofloxacin, norfloxacin, and erythromycin, and was only sensitive to penicillin, amikacin, and tigecycline, to varying degrees. The animal pathogenicity tests showed that the isolate caused feather loss and death in chicks. In summary, *Rothia nasimurium* was isolated from chickens for the first time, and the biological characteristics of the bacterium were investigated in order to provide a reference for the clinical treatment, prevention, and control of *Rothia nasimurium* infection.

**Abstract:**

*Rothia nasimurium* is a facultative anaerobic Gram-positive coccus belonging to the Rothia genus of the *Micrococcaceae* family. While *Rothia nasimurium* is considered an opportunistic pathogen, to date few studies have investigated its pathogenicity and drug resistance. In January 2022, chickens at a poultry farm in China’s Xinjiang Uygur Autonomous Region became ill and died. Treatment with commonly used Chinese medicines and antibiotics was ineffective, causing economic losses to the poultry farm. In order to determine the cause of the disease in these poultry farm chickens, the isolation and identification of the pathogens in the livers and other internal organs of the sick and dead chickens were performed. Further, animal pathogenicity tests, antibiotic susceptibility tests, and the detection of antibiotic resistance genes were carried out to analyze the pathogenicity and drug resistance of the identified pathogens. A Gram-positive coccus was isolated from the livers of the diseased chickens. The isolate was resistant to 17 antibiotics, including ciprofloxacin, chloramphenicol, and florfenicol, and was only sensitive to penicillin, amikacin, and tigecycline, to varying degrees. The results of the drug resistance gene testing indicated that the isolated bacterium carried 13 kinds of resistance genes. Matrix-assisted laser desorption/ionization time-of-flight mass spectrometry, morphological observations, biochemical tests, and 16S rRNA gene sequence analysis were performed on the isolated bacterium, and it was determined that the isolated bacterial strain was *Rothia nasimurium*. The animal pathogenicity tests showed that the isolate caused feather loss and death in chicks; the clinical symptoms and necropsy lesions of the test chicks were consistent with those observed in the farmed chickens. A review of the literature revealed that, to date, there are no reports of infection with *Rothia nasimurium* in chickens. Thus, in this study, *Rothia nasimurium* was isolated from chickens for the first time and an investigation of the biological characteristics of the bacterium was carried out in order to provide a reference for the clinical treatment, prevention, and control of *Rothia nasimurium* infection.

## 1. Introduction

In January 2022, 12-month-old chickens at a poultry farm in Xinjiang Uygur Autonomous Region, China developed a mystery disease. The clinical symptoms observed in the sick chickens included an unstable gait, inactivity, reduced dietary intake, and depression. Some of the weaker chickens sporadically died, with the mortality rate ranging from 0.50 to 2.50%. Dissection of the deceased chickens showed hepatomegaly. The farmer treated the sick chickens with commonly used Chinese medicines, including Jingfang Baidu San, Siwei Chuanxinlian San, and SHUANGHUANGLIAN ZHUSHEYE. Jingfang Baidu San is based on natural ingredients such as *Schizonepeta tenuifolia* Brip, Saposhnikoviae Radix, and Bupleurum, etc. Siwei Chuanxinlian San is based on natural ingredients such as *Andrographis paniculata*, *Isatidis folium*, *Polygonum hydropiper*, and *Tadehagi triquetrum*. SHUANGHUANGLIAN ZHUSHEYE is based on natural ingredients such as *Lonicera japonica* Thunb, Scutellaria Baicalensis, and *Forsythia suspensa*. After these treatments were ineffective, the farmer sought guidance from a veterinarian. Subsequently, the chickens were treated with antibiotics, including gentamicin and ciprofloxacin. However, the condition of the sick chickens did not significantly improve, resulting in economic losses to the poultry farm. In order to determine the cause of the disease in the poultry farm chickens, a dead chicken and a sick chicken with obvious symptoms were randomly selected and brought back to our laboratory for isolation and identification of the infecting pathogen. A series of laboratory tests were performed, and it was identified that the pathogen causing the morbidity and mortality in the poultry farm chickens was *Rothia nasimurium*. Until now, there have been no reports of *Rothia nasimurium* infection in chickens.

*Rothia nasimurium* belongs to the *Rothia* genus of the *Micrococcaceae* family. It is Gram-positive, facultatively anaerobic, does not form spores, and is non-motile. *Rothia nasimurium* was first discovered by Collins et al. in 2000 after it was isolated from the nose of a healthy mouse [1], hence the name. Since then, studies of *Rothia nasimurium* infection in animals have been performed around the world. Researchers have isolated *Rothia nasimurium* from the air of a farm [2], the tonsils of dogs, the external auditory canal, and other infected sites [3], the eggs of white-fronted geese [4], and the tonsils of healthy piglets [5]. Antibiotic susceptibility tests have confirmed that this isolate has a multidrug-resistant phenotype and can produce strong synergistic hemolysis with *Staphylococcus* colonies, indicating that *Rothia nasimurium* as a symbiont or opportunistic pathogen produces pathogenicity in mixed infection with *Staphylococcus*. Since 2021, researchers have isolated *Rothia nasimurium* from ducks [6], rabbits, and geese [7,8], and have found that this bacterial strain can cause severe feather loss in goslings, suggesting that infection can pose a potential disease threat to other hosts. Wang et al. [6] found that *Rothia nasimurium* isolated from ducks in eastern China most likely originated from Amazonian symbionts in Florida, USA. This indicates that *Rothia nasimurium* poses a cross-regional public health infection risk and a potential threat to humans and animals. Therefore, *Rothia nasimurium* has important implications for public health and should be assessed for its risk of global spread.

*Rothia nasimurium* is not a common pathogen in clinical practice, with no reports of human infection with this bacterium to date. However, according to the limited reports on the pathogenicity and drug resistance of *Rothia nasimurium*, it often shows high levels of resistance to a variety of commonly used antibiotics [2,6,8]. Thus, *Rothia nasimurium* deserves the attention of researchers. Studies have shown that some proteins directly involved in the drug resistance of the bacterium originated from non-pathogenic bacteria, such as the efflux pump protein LmrP from *Lactococcus lactis* and BmrA protein from *Bacillus subtilis* [9,10]. In addition, drug resistance can be spread among the same bacterium or different bacteria through plasmids, chromosomes, or drug resistance genes, and even between prokaryotes and eukaryotes [2]. This can result in the widespread distribution of drug-resistant genes and bacteria. Thus, this problem is complex with potential threats to both animals and humans [11,12]. As such, it is particularly important to investigate the drug resistance and drug resistance mechanism of *Rothia nasimurium*. A comprehensive understanding of the drug resistance mechanism will provide new targets for the development of novel antibacterial drugs and potential clinical drug combinations.

In this study, *Rothia nasimurium* was found to have certain pathogenicity in chickens, causing disease and even death. Therefore, a preliminary analysis of the biological characteristics of the bacterium was performed in order to provide a scientific basis for the prevention, control, and rational drug treatment of this bacterial strain so as to reduce its potential harm to the breeding industry and public health.

## 2. Materials and Methods

### 2.1. Isolation and Purification

Two clinically ill 12-month-old chickens were randomly selected from a poultry farm in China’s Xinjiang Uygur Autonomous Region. The chickens exhibited an unsteady gait, inactivity, and reduced dietary intake. The chickens were placed on a sterile test bench and their abdominal cavities were opened. The livers of the sick chickens were found to be enlarged. The chickens were dissected according to standard laboratory methods [13]. An autoclaved cotton swab was dipped deep in the surface of the diseased liver, and the swab was placed in a 2 mL centrifuge tube containing 0.9% normal saline. After standing for 2 h, the tube was fully vortexed and 0.2 mL of liquid was drawn. The liquid was inoculated on blood agar medium (Haibo Biotechnology Co., Ltd, Qingdao, China) and then incubated in a 37 °C incubator for 12 h [14,15]. Then, a single colony was selected, and the single colony was inoculated on a new blood agar plate for further purification. Cultivation was performed overnight at 37 °C [16,17]. A purified bacterial colony was ultimately obtained.

### 2.2. Virus Detection

The liver, lung, spleen, and tracheal tissue samples of the diseased chickens were ground with sterile saline at a ratio of 1:4 in a 2 mL centrifuge tube. The diseased organ solution was frozen (−20 °C) and thawed three times. Then, it was centrifuged at 6000× *g* r/min for 15 min, and the supernatant was obtained. The total virus RNA was extracted with an RNA extraction kit (Tiangen Biotechnology, Beijing, China). The extracted total RNA was used as a template to detect Marek’s disease virus (MDV), infectious laryngotracheitis virus (ILTV), and fowl poxvirus (FPV) using a real-time fluorescence quantitative PCR detection kit (Biolab, Beijing, China). PCR amplification was performed in a 20 μL system containing 2× Probe qPCR MagicMix 10 μL. The probe method was as follows: qPCR Primer Mix 2 μL, qPCR probe 1 μL, DNA template 7 μL. The amplification procedure was as follows: MDV and ILTV: pre-denaturation at 95 °C for 5 min; PCR reaction at 95 °C for 15 s, 60 °C for 1 min, 45 cycles. FPV: pre-denaturation at 95 °C for 3 min; PCR reaction at 95 °C for 15 s, 60 °C for 1 min, 40 cycles. The test results were judged according to the instructions.

### 2.3. Bacterial Identification

#### 2.3.1. Observation of Gram Staining

The pure-cultured bacterial solution was evenly spread on a glass slide, fixed with a flame after natural drying, and then submitted to Gram staining [18]. The slide was examined under an oil immersion microscope to observe the bacterial staining and morphology.

#### 2.3.2. Bacterial Biochemical Test

A 2–10 μL pure-cultured bacterial solution was aseptically aspirated. Then, it was injected into an identification biochemical tube (Hangzhou Binhe Microbial Reagent Co., Ltd., Hangzhou, China) and cultured at 37 °C for 24 h, according to the manufacturer’s instructions. The strain was cultured for 72 h to perform biochemical tests. *Bergey’s Manual of Determinative Bacteriology*, Eighth Edition was consulted for interpretation of the results [19].

#### 2.3.3. Matrix-Assisted Laser Desorption/Ionization Time-of-Flight Mass Spectrometry (MALDI-TOF MS)

Bacterial identification was performed using a Bruker mass spectrometer (Billerica, MA, USA) calibrated using the Brucker bacterial test standard. The Brucker MBT_FC.par method was used for the automatic collection of the data [20,21].

### 2.4. Sequence Analysis of 16S rRNA

According to the instructions of the TIANamp Bacterial DNA extraction kit, pure-cultured bacterial solution was used to extract bacterial DNA (Tiangen Biotechnology, Beijing, China). A microplate reader was used to measure the DNA concentration and OD260/280 of the extracted bacterium. Readings with OD260/280 values of 1.8 to 2.0 were considered qualified. This means that the DNA is of good quality and high purity, with a low content of impurities such as RNA, proteins, polysaccharides, and phenols. This level of purity meets the requirements of conventional PCR-based experiments. The universal primers for bacterial 16S rRNA were synthesized by Shanghai Sangon Bioengineering Technology Service Co., Ltd. (Sangon Biotech, Shanghai, China). The primer sequence was 27F:5′-AGAGTTTGATCATGGCTCAG-3′; 1 492R:5′-TACGGTTACCTTGTTACGACTT-3′, and the expected amplified fragment length was 1500 bp. PCR amplification was performed in a 50 μL system: ddH_2_O 18 μL, 2× PCRMix 25 μL, upstream and downstream primers (10 pmol/μL) 2 μL each, and DNA template 3 μL. The amplification procedure was as follows: pre-denaturation at 95 °C for 5 min; denaturation at 94 °C for 30 s, annealing at 56 °C for 30 s, extension at 72 °C for 2 min, 35 cycles; extension at 72 °C for 10 min. Then, 5 μL of PCR amplification product was aspirated and detected by 1% agarose gel electrophoresis. After gel imaging was performed, the correctly identified and positive amplification products were sent to Shanghai Sangon Bioengineering Technology Service Co., Ltd. for sequence determination. The sequencing results of the isolated strain were compared and analyzed for homology with the NCBI GenBank database. Sequences were compared using the Clustal W method of the MEGA 11 software (Mega Limited, Auckland, New Zealand), and the comparison results were used to build a phylogenetic tree using the minimum evolution method. Phylogenetic analysis was performed with 1000 calibrations to increase the reliability of the analysis.

### 2.5. Antibiotic Susceptibility Testing

Twenty kinds of antibiotics, namely, penicillin, ceftazidime, ampicillin/sulbactam, levofloxacin, cefoxitin, tobramycin, gentamicin, amikacin, tetracycline, tigecycline, ciprofloxacin, norfloxacin, erythromycin, meropenem, imipenem, azithromycin, chloramphenicol, moxifloxacin, clindamycin, and sulfamethoxazole trimethoprim were purchased from Hangzhou Tianhe Microbial Reagent Co., Ltd. (Tianhe Microbial Reagent Co., Ltd, Hangzhou, China) and Shanghai Macklin Biochemical Technology Co., Ltd. (Shanghai Macklin Biochemical Technology Co., Ltd, Shanghai, China). Minimum inhibitory concentrations (MICs) and zone diameters of the isolates were determined separately using a broth microdilution method and a K-B method [22,23] in accordance with the Clinical and Laboratory Standards Institute (CLSI) criteria. (1) The antibiotic sensitivity testing using the broth microdilution method was performed in 96-well plates (Meihuayl, Zhuhai, China); (2) the pure-cultured bacterium solution (0.5 McFarland (MCF) units) was coated on a Mueller–Hinton plate purchased from Haibo Biotechnology Co., Ltd. (Haibo Biotechnology Co., Ltd, Qingdao, China). After drying, the drug-containing paper was pasted, and the diameter of the inhibition zone was measured after overnight culture at 37 °C. The test results were judged according to the CLSI criteria [24,25]. Since no specific antibiotic breakpoints are currently available for *Rothia nasimurium*, clinical resistance breakpoints for closely related Gram-positive anaerobic species and *Staphylococcus* spp. breakpoints were used as a reference. In addition, since Tigecycline does not yet have a CLSI-recommended susceptibility sensitivity breakpoint standard, the FDA-recommended breakpoint of *Staphylococcus* spp. was used as a reference.

### 2.6. Detection of Antibiotic Resistance Genes

Bacterial genomic DNA extraction kits were used to extract the total bacterial genomic DNA as a PCR reaction template. DNA was stored at −20 °C for later use. The detection of 8 resistance genes in 28 categories was carried out in the isolated bacterium, including the β-lactam resistance genes *bla TEM*, *bla SHV*, *bla CTX-M*, *mecA*, *TEM*; sulfonamide sulfonamide resistance genes *sul1*, *sul2*, *sul3*; quinolone resistance genes *aac(6′)-Ib*, *oqxAB*, *qnrA*, *gyrA*, *gyrB*, *parC*, *parE*; aminoglycoside resistance genes *aadB*, *aacC2*, *aph(3′)-Ia*, *aac(6′)/aph(2″)*, *ant(6)-I*; tetracycline resistance genes *tet(A)*, *tet(B)*, *tet(C)*, *tetM*; macrolide resistance genes *ermB*, *mefA*; lincosamide resistance gene *LinA*; and chloramphenicol resistance gene *fexA*. The primers were synthesized and provided by Shanghai Sangon Bioengineering Technology Service Co., Ltd. The primer sequence information is shown in Table 1. PCR amplification was performed in a 25 μL system: ddH_2_O 10.5 μL, 2× PCRMix 12.5 μL, upstream and downstream primers (10 pmol/μL) 0.5 μL each, and DNA template 1 μL. The amplification procedure was as follows: pre-denaturation at 94 °C for 5 min; denaturation at 94 °C for 30 s, annealing temperature at (Table 1) for 30 s, extension at 72 °C for 50 s, 30 cycles; extension at 72 °C for 10 min. Then, 7 μL of PCR amplification product was aspirated and detected by 1% agarose gel electrophoresis.

### 2.7. Animal Pathogenicity Test

Twenty-one healthy one-day-old chicks with strong vitality were purchased from Weidong Hatchery and were numbered 1–21. Before the experiment, the chicks were acclimated to the laboratory environment for three days; the animals had free access to food and water. The experimental methods conformed to the standard operating procedures for animal experiments. First, 100 μL of the bacterial strain in the logarithmic growth phase was obtained and inoculated in brain–heart infusion liquid medium. Then, incubation was performed at 37 °C for 24 h to obtain the test stock solution. The test stock solution was diluted with physiological saline to 3 × 10^8^ CFU/mL by McFarland turbidimetry as backup bacterial fluid. Then, the standby stock solution was diluted to the concentration required for each group of experiments. According to the method of Kang et al. [8], infection in experimental animals was induced orally and via intraperitoneal injection; see Table 2 for the details of each group. The 21 chicks were randomly divided into 7 groups (groups 1–6 were the experimental groups; group 7 was the control group), and each group contained 3 chicks. Each chick was inoculated with 0.2 mL of pure-cultured bacterial solution diluted to the required concentration, and the chicks were observed once every six hours. From days 1 to 7, the mental states of the infected chicks were observed, and appearance changes and clinical symptoms were recorded. The experimental group chicks and the control group chicks were housed separately under the same rearing conditions, with access to clean water and food. When the chicks showed abnormalities, their livers were removed, and Gram staining and bacterial isolation were performed. This study was approved by the Institutional Review Board of Shihezi University (approval number: A2022-67). The animal experiments were carried out in accordance with the relevant requirements of ARRIVE. After the experiment, all experimental chicks were sacrificed by injection of sodium pentobarbital (150 mg/kg, IP) and sent to the experimental animal incineration center.

## 3. Results

### 3.1. Virus Detection Results

The real-time fluorescence quantitative PCR detection results were negative for MDV, ILTV, and FPV.

### 3.2. Isolation and Purification

This study was based on a typical poor Chinese poultry farm. The poultry farm housed the chickens on flat ground and adopted a free-range farming mode. The breeding site was in the woods, surrounded by fences; this type of rearing environment is very open, and the chickens were able to move freely and eat and drink independently. Although the farm had well-established biosecurity measures and routine vaccinations were performed on time, the breeding and management practices of the poultry farm were poor, with irregular cleaning and disinfection. After infection, three to five or more chickens huddled together and exhibited an unsteady gait, lethargy, and reduced feed intake. After the isolated bacterium was cultured in sheep blood agar solid medium for 24 h, the colonies were observed to be round, grey-white in colour, with smooth and neat edges. (Figure 1).

### 3.3. Animal Regression Experiment

#### 3.3.1. Gram Staining

The bacterium was observed under a microscope and was found to be a uniform ellipsoid sphere arranged in a grape-like shape, with a diameter of ≥1.0 μm. Gram staining was positive (Figure 2).

#### 3.3.2. Bacterial Biochemical Test

The results of the biochemical tests indicated that the isolate could only ferment glucose, maltose, and sucrose, but could not ferment lactose, did not produce hydrogen sulfide, could not utilize mannitol and sodium citrate, and showed a negative urease reaction (Table 3). According to *Bergey’s Manual of Determinative Bacteriology*, Eighth Edition, together with the relevant literature, it was determined that the isolated bacterium conformed to the biochemical identification characteristics of *Rothia nasimurium*.

#### 3.3.3. MALDI-TOF MS Identification

The mass spectrum data of the isolated bacterium were compared with the standard database with BrukerBiotyper 3.0 software (Bruker Daltonics, Billerica, Germany). The results produced a score of 2.230, and the isolated bacterial strain was identified as *Rothia nasimurium*.

### 3.4. Alignment of Bacterial 16S rRNA Sequences and Construction of Phylogenetic Tree

The DNA of the strain was extracted, and the OD260/280 value of the microplate reader was between 1.8 and 2.0. The PCR amplification products were detected by 1% agarose gel electrophoresis, and the results showed that the amplified bands were all about 1500 bp (Figure 3; the original western blot figure in Appendix A). The bands were clear and bright and were of the expected size. The sequencing results showed that the similarity between the four isolates and *Rothia nasimurium* was more than 99.5%. Genetic evolution analysis showed that the test bacterium clustered on the same clade with *Rothia nasimurium* (Figure 4). Thus, together, the morphological characteristics of the isolated bacterium, physiological and biochemical characteristics of the isolated bacterium, MALDI-TOFMS identification results, and 16S rRNA gene sequence comparison results indicated that the isolated bacterium was *Rothia nasimurium*. The GenBank accession number of the 16S rRNA sequence of the isolated bacterium is OP120784.1.

### 3.5. Antibiotic Susceptibility Testing

Antibiotic susceptibility testing of the isolate was performed with 20 different antibiotics. The results of the drug sensitivity test using the K-B method (Table 4) and the drug sensitivity test using the broth microdilution method (Table 5) were consistent, indicating that, according to the CLSI criteria, the isolate exhibited multidrug resistance, with strong resistance to ciprofloxacin, imipenem, chloramphenicol, and meropenem, and serious resistance to ceftazidime, ampicillin/sulbactam, levofloxacin, cefoxitin, tobramycin, gentamicin, tetracycline, norfloxacin, erythromycin, azithromycin, moxifloxacin, sulfamethoxazole trimethoprim, and clindamycin. The diameter of the inhibition zone was 0. Different degrees of sensitivity to penicillin, amikacin, and tigecycline were observed, among which tigecycline exhibited the highest level of inhibition.

Since tigecycline is not used in the animal industry, penicillin is the most appropriate drug treatment based on the results of antibiotic susceptibility testing. However, in clinical treatment, attention should be paid to the dosage and combination of drugs to reduce the generation of drug-resistant strains. Of note, in order to prevent errors in the test results, three parallel tests were performed for each antibiotic diffusion disk during the testing protocol, and the tests were each repeated once. The test results were the same.

### 3.6. Detection of Antibiotic Resistance Genes

The results of the drug resistance gene testing indicated that the isolated bacterium carried 13 kinds of resistance genes, including the β-lactam resistance genes bla TEM, *bla CTX-M*, *mecA*, *TEM*; sulfonamide resistance genes *sul1*, *sul2*, *sul3*; quinolone resistance genes *aac(6′)-Ib*, *gyrA*; aminoglycoside resistance gene *aph(3′)-Ia*; tetracycline resistance genes *tet(A)*, *tetM*; and macrolide resistance gene *ermB*. The target strip was consistent with the expected size (Figure 5; the original Western blot figure in Appendix A). The other 15 genes were not detected.

### 3.7. Animal Pathogenicity Experiment

Three days after infection, the chicks in the experimental group began to show the same clinical symptoms as the naturally infected chickens (Figure 6A). The symptoms included lethargy, unsteady gait, and huddling together. The chickens also exhibited reduced activity and dietary intake. The backs of the chicks in the intraperitoneal injection group were missing some feathers. It was observed that the higher the concentration of bacterial solution administered via intraperitoneal injection, the more obvious the feather loss in the chicks (Figure 6B). On the fourth day after infection, the clinical symptoms of the chicks in the intraperitoneal injection group were aggravated, and feather loss was more obvious than the previous day; again, the higher the concentration of injected bacterial solution, the more obvious the feather loss (Figure 6C). Mild feather loss was also observed in the oral administration group. Similarly, the higher the concentration of oral bacterial solution, the more obvious the feather loss in the chicks (Figure 6D). Five days after infection, the 3 × 10^8^ CFU chicks in the intraperitoneal injection group died and suffered severe feather loss (Figure 6E). The dead chicks and the symptomatic chicks were sacrificed, and necropsy revealed that the livers were mildly enlarged. Hyperemia and hemorrhage were also observed, and a spherical bacterium was found by Swiss staining of liver tissue sections (Figure 6F). The bacterium in the liver, spleen, and blood was isolated and cultured. The colony morphology, Gram staining, and biochemical test results were consistent with the original isolates, and the 16S rRNA test results also showed that the isolated bacterium was *Rothia nasimurium* causing the morbidity and mortality in the farming chickens. The chicks in the control group showed no changes post-inoculation, and no bacterium was isolated from the liver, spleen, or other parenchymal organs.

## 4. Discussion

Since *Rothia nasimurium* is not a common pathogen in clinical practice, there are few reports of *Rothia nasimurium* infection. In addition, *Rothia nasimurium* is considered an opportunistic pathogen and is part of the normal flora of animals. Thus, to date, this bacterium has not attracted significant research attention. However, due to its multidrug resistance and pathogenicity, researchers have begun to pay more attention to it in recent years [5,6,7,8].

In this study, a bacterial strain was isolated from the livers of sick chickens that died during the course of clinical disease. The bacterial strain was identified as *Rothia nasimurium*. The results of antibiotic susceptibility testing showed that the isolate was multidrug resistant and only sensitive to tigecycline, penicillin, and amikacin. This is consistent with the results of the studies of Wang et al. [6] and Kang et al. [8]. However, the isolated strain showed resistance to ampicillin/sulbactam, which is different to the results of Kang et al. [8]. This may be due to different medication habits in different regions. With the increasing abuse of antibiotics, the problem of bacterial resistance has become a global issue and a hotspot in scientific research. The emergence of antimicrobial resistance poses a huge threat to public and animal health, especially in less developed countries where animals used for food often mix with humans [26]. Studies have also shown that some drug resistance genes are located on bacterial plasmids, transposons, integrons, and other genetic materials, and bacteria can transfer these movable genetic elements between species through transformation, transduction, conjugation, etc. The transmission of drug-resistant genes leads to the spread of drug-resistant genes among different species of bacteria, resulting in the wide spread of drug-resistant strains. Wang et al. [6] reported the possible dissemination of resistant genes from the multidrug-resistant commensal bacteria of wild birds (Amazon parrots) to *Rothia nasimurium* in the bacterial flora of other animals in different environments, which may lead to the emergence of these genes in livestock, poultry, and even humans. To control the spread of drug-resistant genes, on the one hand it is necessary to standardize the use of antibiotics, and on the other hand it is necessary to effectively control the transfer and spread of drug-resistant genes. Research has also shown that there is a risk of bacteria spreading across regions. Therefore, the potential public health risk of the multidrug-resistant strain *Rothia nasimurium* should be taken seriously. To date, the number of high-level drug-resistant bacteria globally has shown an increasing trend and includes clinically common pathogens such as *Staphylococcus aureus* [27], *Acinetobacter baumannii*, and *Pseudomonas aeruginosa* [28]. The superbug, methicillin-resistant *Staphylococcus* (MRSA), is now widely reported around the world [29,30]. In addition, while the number of drug-resistant bacteria is increasing, drug resistance is also increasing [31,32]. However, there are few reports on the high-level drug resistance of *Rothia nasimurium*. A search of the literature identified only a few related articles [2,6,8].

As an opportunistic pathogenic bacterium, the occurrence and spread of *Rothia nasimurium* are often directly related to a lack of attention to animal welfare by chicken farmers, inappropriate feeding management methods, unsanitary conditions, and inadequate epidemic prevention systems. Due to limited professional knowledge, most free-range chicken farmers do not pay much attention to disease prevention and control. Free-range chickens often do not have an appropriate coop or clean eating environment; these chickens live for a long time under conditions conducive to the rampant reproduction of pathogenic microorganisms, which can easily lead to diseases in the chickens. In terms of disease prevention and control on farms, it is first necessary to strengthen daily feeding management, pay attention to the balanced combination of various nutrients, improve the immune function of the animals, and strengthen the chickens’ resistance to pathogens. Second, biosafety prevention and control measures, such as sanitation and disinfection, should be implemented to mitigate the transmission of bacterial diseases. Third, attention should be paid to the control of breeding density. A reasonable breeding density can not only reduce the occurrence of diseases but also provides a good environment for the healthy growth of chickens. Fourth, when diseases occur in chicken flocks, scientific and reasonable treatment of sick and dead chickens should be adopted to eliminate the transmission of the pathogen. When choosing a therapeutic drug, it is necessary to select a drug that is sensitive to the bacterial strain [33,34]. This can not only effectively control the disease and reduce the production of a super bacterium but can also prevent an imbalance of gut flora in animals, which can occur due to blind drug use. The antibiotic susceptibility test results in this study showed that the best drug treatment for *Rothia nasimurium* is penicillin. This result provides a scientific basis for rational drug use to effectively mitigate bacterial resistance [35,36]. In addition, in terms of existing treatments, the therapeutic efficiency of targeted drugs can be improved by means of combined medication while paying attention to the administration method and dosage. Moreover, in the context of the increasing cross-species transmission of high-level drug-resistant strains, it is necessary to recognize the significant potential threat of high-level drug-resistant pathogenic microorganisms to the breeding industry; therefore, effective monitoring and control mechanisms must be established so as to ensure the safety of animal husbandry production and public health. At face value, protecting animal welfare restricts the freedom of humans and the rights of humans to use animals. However, from the perspective of development and connection, it is necessary to protect animal health and produce safe and nutritious animal products, thus protecting human safety and health. To move towards civilization, human beings should not only care about the relationships between people but, more importantly, care about the relationships between people and all living things.

As a class of environmental pollutants, drug resistance genes have received increasing attention. The World Health Organization reported that antibiotics resistance genes will be one of the most significant challenges to human health in this century. The BBC reported that the death toll from antibiotic-resistant infection would exceed that of cancer by 2050 [37]. Human exposure to veterinary antibiotics (VA) and preferred as veterinary antibiotics (PVAs) via the food chain is unavoidable due to their extensive use not only for treating bacterial infections but also as growth promoters in livestock and aquaculture [38]. This leads to the generation and storage of resistance genes in animals consumed for food. These resistance genes not only directly affect the prevention and control of foodborne animal diseases but also spread along the food chain, ultimately endangering food and public health [39]. Therefore, it is important to test the resistance genes of drug-resistant bacteria in animals consumed for food. In this study, a total of 13 resistance genes were detected: *bla TEM*, *bla CTX-M*, *mecA*, *TEM*, *sul1*, *sul2*, *sul3*, *aac(6′)-Ib*, *gyrA*, *aph(3′)-Ia*, *tet(A)*, *tetM*, *ermB*. Wang et al. [6] performed whole genome sequencing of the *Rothia nasimurium* Shandong isolate. Multiple resistance genes were detected, including *aac(6′)-Ib*, *ant(3″)-Ia*, *sul1*, *dfrA7*, and *erm(X)*. At the same time, multidrug resistance active efflux pumps were also detected, including *tetZ*, *cmx*, *pstB*, and *qacEΔ1*. This is consistent with the findings of the current study indicating that *Rothia nasimurium* carries multiple resistance genes. However, no resistance genes for the lincosamide and chloramphenicol classes were detected in this study, and the susceptibility results indicated that the isolate was resistant to chloramphenicol and clindamycin. This suggests the possible presence of other undetected resistance genes or other resistance mechanisms that have not yet been discovered.

The results of the animal pathogenicity experiments indicated that chicks could be infected by intraperitoneal injection and oral inoculation and that the clinical symptoms and necropsy lesions were consistent with those of chickens with naturally occurring *Rothia nasimurium* injection. Further, *Rothia nasimurium* was isolated from the livers and blood of the experimental chicks, indicating that *Rothia nasimurium* can penetrate the immune barrier of the human body and pose a direct threat to human health. The test results also indicated that *Rothia nasimurium* can cause death in chicks, indicating that this bacterial strain has certain pathogenicity in chickens. Therefore, the bacterium has the potential to harm researchers, experimental animals, and human health. Kang et al. [8] found that *Rothia nasimurium* can cause severe feather loss in goslings. In this study, infected chicks also exhibited feather loss, and thus it can be speculated that *Rothia nasimurium* may also have depilatory effects in other poultry species. Therefore, we plan to conduct further studies on the multidrug resistance mechanism of *Rothia nasimurium* and the mechanism underlying feather loss in chicks.

## 5. Conclusions

In this study, a strain of *Rothia nasimurium* was found in sick chickens obtained from a poultry farm in Xinjiang Uygur Autonomous Region. Both intraperitoneal injection and oral inoculation of this bacterial strain caused clinical symptoms such as feather loss in chicks, and an intraperitoneal injection of pure-cultured bacterial solution at a concentration of 3 × 10^8^ CFU caused chicks to die, indicating that this bacterium poses a potential threat to the breeding industry and human health. The results of the antibiotic susceptibility testing showed that the strain was multidrug resistant, with sensitivity only found to tigecycline, penicillin, and amikacin. The results of the drug resistance gene testing indicated that the isolated bacterium carries multiple resistance genes. Due to the multidrug resistance and certain pathogenicity of this bacterium, as well as the risk of cross-regional transmission, future research on this bacterium is a priority, and its biological characteristics and mechanism of action should be further studied. Moreover, further research is required to understand the prevalence and transmission law of this bacterial strain in order to better control its spread. The results of this study provide a reference for the prevention and clinical treatment of *Rothia nasimurium* infection and can serve as a guide for formulating prevention and control measures for avian infection.

## Figures and Tables

**Figure 1 vetsci-09-00653-f001:**
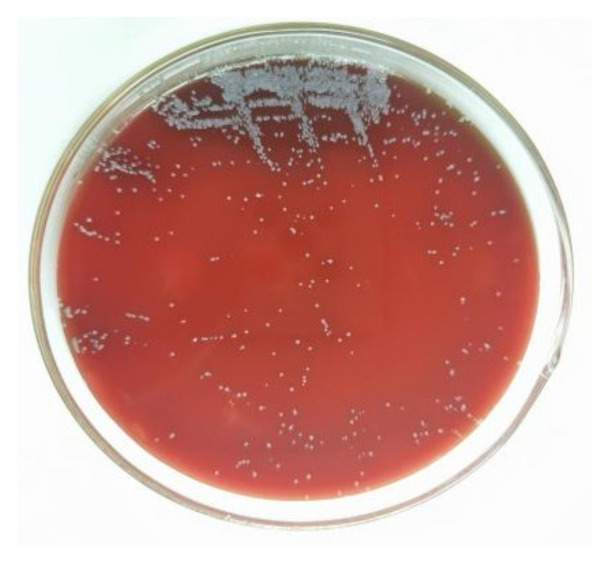
Purified bacterial colonies cultured on blood agar.

**Figure 2 vetsci-09-00653-f002:**
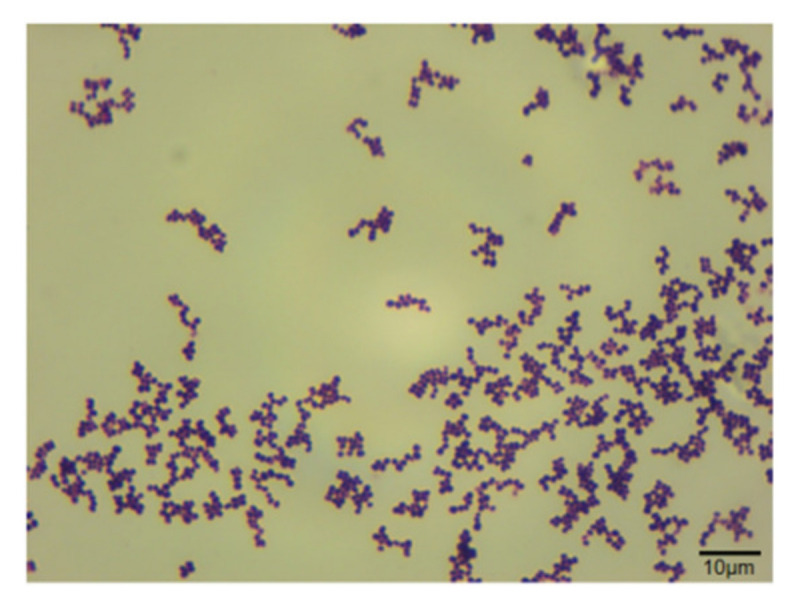
Gram staining of the isolated bacterium (1000× magnification).

**Figure 3 vetsci-09-00653-f003:**
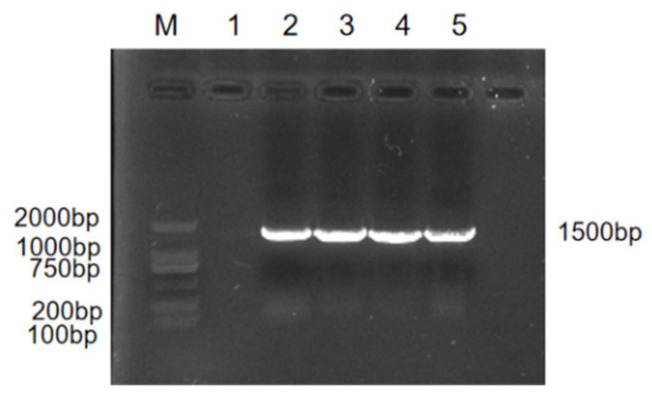
Agarose gel electrophoresis (cropped).

**Figure 4 vetsci-09-00653-f004:**
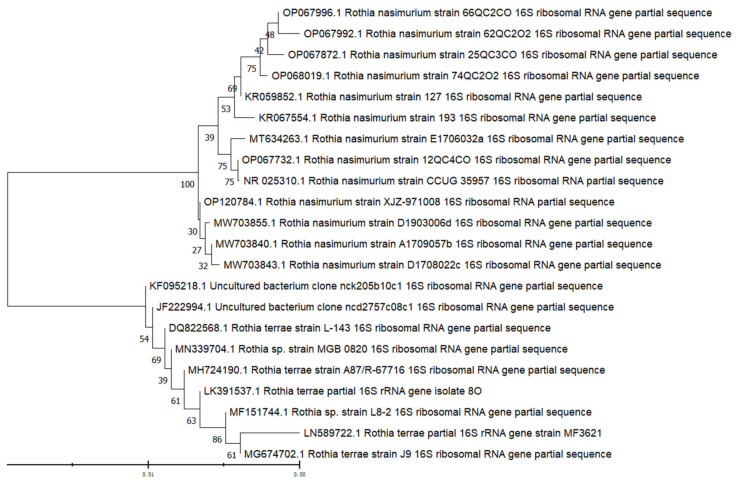
Phylogenetic tree based on bacterial 16S rRNA genes. Lane M: DL-2000 marker; Lane 1: negative control; Lanes 2–5: 16S rRNA amplification results of isolated strains.

**Figure 5 vetsci-09-00653-f005:**
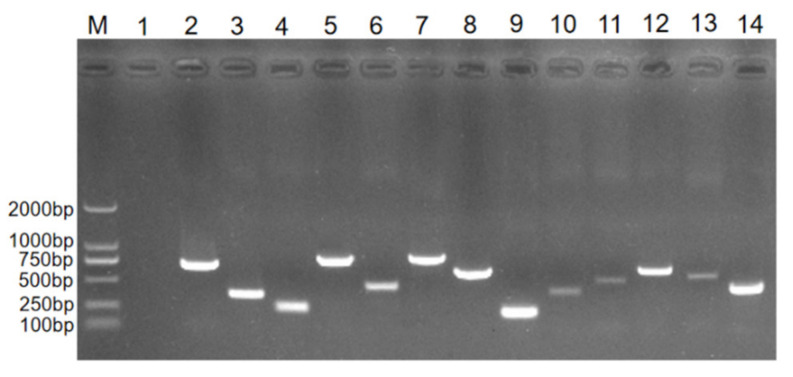
PCR results of drug resistance genes (cropped). Lane M: DL-2000 marker; Lane 1: negative control; Lanes 2–14: drug resistance genes: *bla TEM*, *bla CTX-M*, *sul1*, *sul2*, *sul3*, *gyrA*, *aph(3′)-Ia*, *tet(A)*, *mecA*, *tetM*, *ermB*, *TEM*, *aac(6′)-Ib*.

**Figure 6 vetsci-09-00653-f006:**
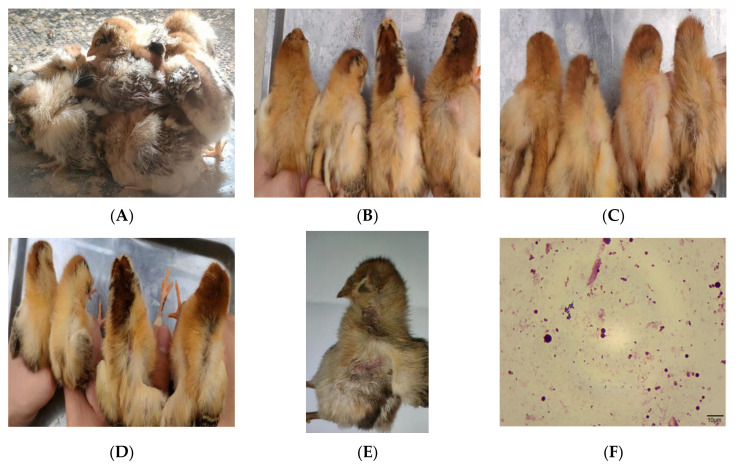
(**A**) The status of the experimental group chicks three days post-infection. (**B**) Three days after infection. Left: feather condition of control group chicks, middle left: 3 × 10^5^ cfu intraperitoneal injection chicks, middle right: 3 × 10^7^ cfu intraperitoneal injection chicks, right: 3 × 10^8^ cfu intraperitoneal injections checks. (**C**) Four days after infection. Left: feather condition of control group chicks, middle left: 3 × 10^5^ cfu i.p. chicks, middle right: 3 × 10^7^ cfu i.p. chicks, right: 3 × 10^8^ cfu i.p. chicks. (**D**) Four days after infection. Left: feather condition of control group chicks, middle left: 3 × 10^5^ cfu-infused chicks, middle right: 3 × 10^7^ cfu-infused chicks, right: 3 × 10^8^ cfu-infused chicks. (**E**) Five days after infection. The chicks in the 3 × 10^8^ cfu intraperitoneal injection group died and severe depilation was observed. (**F**) Swiss staining of liver tissue (1000× magnification).

**Table 1 vetsci-09-00653-t001:** Primer sequence information of drug resistance genes.

Antimicrobial Types	Genes	Primer Sequences (5′→3′)	Annealing Temperature	Product Size/bp
β-lactams	*bla TEM*	F:CAGAAACGCTGGTGAAAGTA R:ACTCCCCGTCGTGTAGATAA	55	719
	*bla SHV*	F:ATGCGTATATTCGCCTGTG R:CCTCATTCAGTTCCGTTTCC	55	502
	*bla CTX-M*	F:AGTGAAAGCGAACCGAATC R:CTGTCACCAATGCTTTACC	55	365
	*mecA*	F:GTTGTAGTTGTCGGGTTTGG R:GTTGTAGTTGTCGGGTTTGG	56	336
	*TEM*	F:AGGAAGAGTATGATTCAACA R:CTCGTCGTTTGGTATGGC	51	535
Sulfonamides	*sul1*	F:CATTGCCTGGTTGCTTCAT F:CATTGCCTGGTTGCTTCAT	54	238
	*sul2*	F:CATCATTTTCGGCATCGTC R:TCTTGCGGTTTCTTTCAGC	54	793
	*sul3*	F:AGATGTGATTGATTTGGGAGC R:TCTTGCGGTTTCTTTCAGC	54	443
Quinolones	*aac(6′)-Ib*	F:TTGCGATGCTCTATGAGTGGCTA R:CTCGAATGCCTGGCGTGTTT	55	482
	*oqxAB*	F:GATCAGTCAGTGGGATAGTTT R:TACTCGGCGTTAACTGATTA	55	671
	*qnrA*	F:TCAGCAAGAGGATTTCTCA R:GGCAGCACTATTACTCCCA	54	627
	*gyrA*	F:GGTGACGTAATCGGTAAATA R:ACCATGGTGCAATGCCACCA	53	810
	*gyrB*	F:CTCCTCCCAGACCAAAGACA R:TCACGACCGATACCACAGCC	59	448
	*parC*	F:GCGAATAAGTTGAGGAAT R:AGCTCGGAATATTTCGAC	55	417
	*parE*	F:CTGAACTGCTGGCGGAGATG R:GCGGTGGCAGTGCGACGTAA	59	483
Aminoglycosides	*aadB,*	F:GAGGAGTTGGACTATGGATT R:CTTCATCGGCATAGTAAAA	53	208
	*aacC2*	F:GCAATAACGGAGGCAATTCGA R:CTCGATGGCGACCGAGCTTCA	56	697
	*aph(3′)-Ia,*	F:ATGGGCTCGCGATAATGTC R:CTCACCGAGGCAGTTCCAT	56	600
	*aac(6′)/aph(2″)*	F:CCAAGAGCAATAAGGGCATA R:CACTATCATAACCACTACCG	56	220
	*ant(6)-I*	F:ACTGGCTTAATCAATTTGGG R:GCCTTTCCGCCACCTCACCG	56	597
Tetracyclines	*Tet(A)*	F:GCTACATCCTGCTTGCCTTC R:CATAGATCGCCGTGAAGAGG	59.5	210
	*tet(B)*	F:TTGGTTAGGGGCAAGTTTTG R:GTAATGGGCCAATAACACCG	59.5	659
	*tet(C)*	F:CTTGAGAGCCTTCAACCCAG R:ATGGTCGTCATCTACCTGCC	59.5	418
	*tetM*	F:GTGTGACGAACTTTACCGAA R:GCTTTGTATCTCCAAGAACAC	52	510
Macrolides	*ermB*	F:GAAAAGGTACTCAACCAAATA R:AGTAACGGTACTTAAATTGTTTAC	48	636
	*mefA*	F:AACTATCATTAATCACTAGTGC R:TTCTTCTGGTACTAAAAGTGG	50	346
Lincosamides	*LinA*	F:GGTGGCTGGGGGGTAGATGTATTAACTGG R:GCTTCTTTTGAAATACATGGTATTTTTCGA	57	323
Chloramphenicols	*fexA*	F:TTGGGAAGAATGGTTCAGGG R:ATCGGCTCAGTAGCATCACG	50	977

**Table 2 vetsci-09-00653-t002:** Results of the animal pathogenicity experiment.

Group	Quantity/pcs	Concentration/(CFU·mL^−1^)	Dosage/mL	Infection Method	Number of Deaths/pcs	Mortality Rate/%
1	3	3 × 10^5^	0.2	orally	0	0
2	3	3 × 10^7^	0.2	orally	0	0
3	3	3 × 10^8^	0.2	orally	0	0
4	3	3 × 10^5^	0.2	injected intraperitoneally	0	0
5	3	3 × 10^7^	0.2	injected intraperitoneally	0	0
6	3	3 × 10^8^	0.2	injected intraperitoneally	3	100
7	3	0.9% normal saline	0.2	injected intraperitoneally	0	0

**Table 3 vetsci-09-00653-t003:** Bacterial biochemical test.

Substrate	Result
Maltose	+
Sucrose	+
Glucose	+
Lactose	−
Urea	−
Mannitol	−
Sodium citrate	−
Hydrogen sulfide	−

Note: “+” means positive reaction; “−” means negative reaction.

**Table 4 vetsci-09-00653-t004:** Results of the drug sensitivity test (K-B method).

Drug Name	Judging Standard	Actual Result	Result	Drug Name	Judging Standard	Actual Result	Result
Penicillin	≥15, ≤14	21	susceptible	Ciprofloxacin	≥21, ≤15	12	resistant
Ceftazidime	≥18, ≤14	0	resistant	Norfloxacin	≥17, ≤12	0	resistant
Ampicillin/sulbactam	≥15, ≤11	0	resistant	Erythromycin	≥23, ≤13	0	resistant
Levofloxacin	≥17, ≤13	0	resistant	Meropenem	≥20, ≤15	13	resistant
Cefoxitin	≥18, ≤14	0	resistant	Imipenem	≥23, ≤19	17	resistant
Tobramycin	≥15, ≤12	0	resistant	Azithromycin	≥13, ≤12	0	resistant
Gentamicin	≥15, ≤12	0	resistant	Chloramphenicol	≥18, ≤12	10	resistant
Amikacin	≥17, ≤14	20	susceptible	Moxifloxacin	≥24, ≤20	0	resistant
Tetracycline	≥15, ≤11	0	resistant	Sulfamethoxazole-trimethoprim	≥16, ≤10	0	resistant
Tigecycline	≥15, ≤13	22	susceptible	Clindamycin	≥21, ≤14	0	resistant

Highly sensitive: the test result was above the maximum value. Mediation: the result of a trial between the maximum and minimum values. Resistance: test results below minimum.

**Table 5 vetsci-09-00653-t005:** Results of the drug sensitivity test (broth microdilution method).

Drug Name	Mic (mg/L)	Actual Result	Drug Name	Mic (mg/L)	Result
Penicillin	<0.12	susceptible	Ciprofloxacin	64	resistant
Ceftazidime	>128	resistant	Norfloxacin	>16	resistant
Ampicillin/sulbactam	32/16	resistant	Erythromycin	>128	resistant
Levofloxacin	>64	resistant	Meropenem	128	resistant
Cefoxitin	>32	resistant	Imipenem	8	resistant
Tobramycin	>16	resistant	Azithromycin	>8	resistant
Gentamicin	>16	resistant	Chloramphenicol	>32	resistant
Amikacin	<16	susceptible	Moxifloxacin	>8	resistant
Tetracycline	>16	resistant	Sulfamethoxazole-trimethoprim	>320	resistant
Tigecycline	<0.5	susceptible	Clindamycin	>128	resistant

Highly sensitive: the test result was above the maximum value. Mediation: the result of a trial between the maximum and minimum values. Resistance: test results below minimum.

## Data Availability

The datasets used during the current study are available from the corresponding author upon reasonable request. The datasets generated during the current study are available in the GenBank repository [40], (https://www.ncbi.nlm.nih.gov/nuccore/OP120784.1, accessed on 2 August 2022).

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
