# Peer review of "Rothia nasimurium as a Cause of Disease: First Isolation from Farmed Chickens"

_vetsci, 2022, doi:10.3390/vetsci9120653_

Round 1

Reviewer 1 Report (Previous Reviewer 2)

Dear Authors,

Most changes were accepted. 

1) Family is Micrococcaceae and not Micrococcus

2) Figure 1: The plate is sheep blood agar???? The colour is too dark, almost like a chocolate agar.

3) Table 3: You missed the column of results for the first batch of antibiotics.

This manuscript is extremely similar with the Kang, Y.H.; Zhou, H.S.;Jin, W.J. Rothia nasimurium as a Cause of Disease: First Isolation from Farmed Geese. Veterinary 511 Sciences. 2022, 9, 5 Even wording and structure. For me eventhough changes were accepted, the similarity is unacceptable.

Author Response

Response to Reviewer 1 Comments

Dear Reviewer,

On behalf of my co-authors, I appreciate you very much for your positive and constructive comments and suggestions on our manuscript entitled“Rothia nasimurium as a Cause of Disease: First Isolation from Farmed Chickens” (ID:vetsci-1897783). The comments are valuable for revising the paper, as well as of important guiding significance to our researches. I have studied comments carefully and have made corresponding correction. Revised portions are made in red in the revised manuscript. Once again, I sincerely thank you for your constructive and positive comments.

I have tried our best to revise the manuscript according to your comments. Attached please find the revised version, which I would like to submit for your kind consideration. The main corrections in the paper and the responses to your comments are as follows:

Point 1: Family is Micrococcaceae and not Micrococcus.

Response 1: Thank you very much for your valuable comments. Thank you so much for your careful check. I am very sorry for my negligence. Follow your comments, I have corrected the typo and changed Micrococcus to Micrococcaceae, as shown in Lines 27 and 70.

Point 2: Figure 1: The plate is sheep blood agar???? The colour is too dark, almost like a chocolate agar.

Response 2: Thank you very much for your valuable comments. Follow your comments, I have replaced the new purified bacterial colonies cultured picture, as shown in Figure 1. In order to make more differences with the content of Kang et al.[1], I was wondering if I could replace sheep blood agar with BHI solid medium? The figure below shows the purified bacterial colonies cultured on BHI solid medium and new sheep blood agar, and all have been sent to Editorial office in the form of attachments. Which picture do you think is more appropriate? I will respect and take your comments to make changes to the manuscript. I sincerely hope that my correction will meet with your approval. 

References

Kang, Y.H.; Zhou, H.S.; Jin, W.J. Rothia nasimurium as a Cause of Disease: First Isolation from Farmed Geese. Veterinary Sciences. 2022, 9, 197.

BHI solid medium           sheep blood agar

Point 3: Table 3: You missed the column of results for the first batch of antibiotics.

Response 3: Thank you very much for your valuable comments. Thank you so much for your careful check. I am very sorry for my negligence. Follow your comments, I have replenished the column of results for the first batch of antibiotics, as shown in Table 3.

Point 4: This manuscript is extremely similar with the Kang, Y.H.; Zhou, H.S.;Jin, W.J. Rothia nasimurium as a Cause of Disease: First Isolation from Farmed Geese. Veterinary 511 Sciences. 2022, 9, 5 Even wording and structure. For me eventhough changes were accepted, the similarity is unacceptable.

Response 4: Thank you very much for your valuable comments. First of all, I would like to express my gratitude to the professor for his recognition and acceptance of this manuscript. As we all know, Veterinary Sciences is an excellent journal in this field, with great influence, and the articles that can be published in Veterinary Sciences are reviewed by excellent reviewers, which is innovative and meaningful. Kang et al.[2] found that a strain of Rothia nasimurium was able to survive and cause disease in geese, and it could also cause serious depilation of goslings, they believe this indicates that Rothia nasimurium has potential to harm geese or other animals. This article can be published in veterinary science, and in addition to the content of the article, I think the wording and structure are equally excellent and worth learning. In addition, since our research is similar to the content and direction of the above research, some places inevitably look similar, and I sincerely hope to receive your support and understanding. In terms of research content, our study has made several improvements based on Kang et al.[2]. First, our study found that Rothia nasimurium could survive and cause disease in chickens and could cause serious depilation of chicks, validating the hypothesis of Kang et al.[2]. In addition, a 3×108 CFU/mL test group was added to the Animal pathogenicity test, and it was found that this concentration could cause chick death after 5 days of infection by intraperitoneal injection. Second, Academic Editor believe that the K-B method is not sufficient for antibiotic resistance study. Follow academic editor’s valuable comments, I tested MICs as recommended by CLSI. Third, Academic Editor believe that the authors should investigate the resistance mechanisms, such as some known resistance genes for the antibiotics tested in this study. That would make the paper fit to the topic of this Special Issue. Follow academic editor’s valuable comments, I detected isolated bacterium for antibiotic resistance genes. For a good article, I think its research content is more important than its wording and structure. I sincerely hope that my explanation can be approved by you. 

References

Kang, Y.H.; Zhou, H.S.; Jin, W.J. Rothia nasimurium as a Cause of Disease: First Isolation from Farmed Geese. Veterinary Sciences. 2022, 9, 197.

In addition, I have corrected carefully the several minor mistakes and inaccurately expressed, as shown in Lines 98, 109, 440, 445, 462, 465, 468, 529, 552, 623.

I sincerely thank you and the reviewers for their enthusiastic work, and desire that the correction will meet with your approval. Once again, I would like to thank the referee again for taking the time to review our manuscript.

Thank you and best regards.

Yours sincerely

author:

Jiahao Zhang

Name:lianrui Li

Reviewer 2 Report (New Reviewer)

The authors described an outbreak of Rothia nasimurium in a Chine chicken farm, for the first time. The presence of this microorganism poses a welfare concern and a public health risk, so I understand the importance of the report. A few details could be improved:

- No additional samples on chicken on the farm were collected to confirm the origin of outbreak (Rothia was only isolated from 2 chickens which seems very low to confirm the outbreak and all the remaining work was performed experimentally); to be in fact possible to confirm the origin of the outbreak, further samples should be collected;

- the authors refer several times animals were treated by common chinese medicines. What do they refer to? Is this acceptable as a veterinarian practice? Are the effects of these medicines experimentally confirmed?

- Rothia was confirmed in livers, lungs and spleens and other organs, but no information is provided regarding which other organs;

- Little information is also provided regarding the methods, for example bacterial genomic extraction kits were used but there is no information on which were used;

- on section 2.7, the authors refer how animals were monitored and refer the number of deaths as an outcome. This is not acceptable. Were humane endpoints applied?

- Chickens were inoculated orally and intraperitoneally? What is the relevance of this second route as this is unlikely to be the cause of the natural infection in farms? Did it provided any useful information for disease control or treatment?

- Focus on the causes of the outbreak could be higher as possibly low welfare levels were at the origin of the outbreak as this is an opportunistic disease. The impact that poor housing conditions may have on animal welfare was not discussed and possible treatments with antibiotics, that should be avoided, were the main focus. Not only sanitation is important but also animal density and the conditions how animals are housed

- Recommendations for non terminal sampling could be given or to prevent outbreaks of this disease could have been provided.

- Very little information about the housing conditions at the farm and at the lab were provided

Author Response

Response to Reviewer 2 Comments

Dear Reviewer,

On behalf of my co-authors, I appreciate you very much for your positive and constructive comments and suggestions on our manuscript entitled“Rothia nasimurium as a Cause of Disease: First Isolation from Farmed Chickens” (ID:vetsci-1897783). The comments are valuable for revising the paper, as well as of important guiding significance to our researches. I have studied comments carefully and have made corresponding correction. Revised portions are made in red in the revised manuscript. Once again, I sincerely thank you for your constructive and positive comments.

I have tried our best to revise the manuscript according to your comments. Attached please find the revised version, which I would like to submit for your kind consideration. The main corrections in the paper and the responses to your comments are as follows:

Point 1: No additional samples on chicken on the farm were collected to confirm the origin of outbreak (Rothia was only isolated from 2 chickens which seems very low to confirm the outbreak and all the remaining work was performed experimentally); to be in fact possible to confirm the origin of the outbreak, further samples should be collected;

Response 1: Thank you very much for your valuable comments. Sorry professor, my description here is inaccurate. I feel sorry for the confusion caused to you. As described in the manuscript, this study was based on a typical poor Chinese poultry farm. The poor poultry farm is small at only about 140 chickens. When we investigated the cause of the disease in these poultry farm chickens, only 3 sick chickens died in the poultry farm, and 5 chickens survived clinically ill, and we brought both sick chickens and dead chickens back to the laboratory for testing, and Rothia nasimurium was detected in all chickens. Due to the small number of sick chickens, I don't think it can be called an outbreak. As for your comment to collect more samples to confirm the source of the origin of the outbreak, this is a good proposal that will help prevent and control the bacterial disease and has important clinical guidance value. We're sorry, but we didn't take this work into account before. Due to the epidemic prevention policies of our cities and schools, we are unable to leave the school and nothing from the outside can come in. Therefore, unfortunately we are unable to complete this work, please forgive and understand. I would like to express my sincere gratitude for your enthusiastic work and professional rigor, and please ask the professor to consider my appeal。

Point 2: the authors refer several times animals were treated by common chinese medicines. What do they refer to? Is this acceptable as a veterinarian practice? Are the effects of these medicines experimentally confirmed?

Response 2: Thank you very much for your valuable comments. The Chinese medicines mentioned in the manuscript include Jingfang Baidu San, Siwei Chuanxinlian San and SHUANGHUANGLIAN ZHUSHEYE. Yes, this is an acceptable veterinarian practice. With the development of intensive production in modern poultry farming, antibiotics, synthetic antibacterial drugs and biological vaccines have been widely used in avian disease prevention and breeding production, resulting in increasingly serious bacterial drug resistance. Traditional Chinese medicine is an important part of the splendid culture of the Chinese nation, which has endured for thousands of years and has become the common wealth of the treasure house of human medicine. Some traditional Chinese medicines and prescriptions have achieved good results in the treatment of bacterial infectious diseases, and a large number of in vitro experimental studies have also confirmed that some traditional Chinese medicines have good antibacterial effects on common drug-resistant bacteria. Therefore, the treatment of bacterial infectious diseases with traditional Chinese medicine has attracted more and more attention from researchers [1-5]. Traditional Chinese medicine is rich in resources and relatively low in price, it not only has antibacterial, bactericidal and antiviral effects, but also can regulate and promote the body's immune function, and has non-specific anti-pathogenic microbial effects. As a natural product, traditional Chinese medicine has low residual toxins or non-toxicity, few side effects, and is not easy to develop drug resistance, and has a very broad application prospect in the clinical treatment of infections caused by multidrug-resistant bacteria.

  • Jingfang Baidu San has outstanding application in veterinary clinic, the role is anti-virus, improve resistance, mainly prevent wind cold, wind heat cold and viral diseases. It is widely used in a variety of animal infectious diseases and has remarkable efficacy, including bovine flu, horse flu, Newcastle disease, mild avian influenza, chicken viral respiratory disease, duckling viral hepatitis and other diseases[5], and is regarded as the first-line medicine for veterinary epidemic prevention.
  • Siwei Chuanxinlian San is currently one of the most important Chinese veterinary medicines, is a classic formula commonly used in veterinary clinical practice, and is now included in the Second Department of the Veterinary Pharmacopoeia of People's Republic of China (2010 Edition). The medicine is composed of 4 herbs of andrographis, large green leaf, spicy thistle and gourd tea, which has the effect of clearing heat and detoxification, dehumidification and stagnation. Andrographis has antibacterial, antiviral, anti-inflammatory, antitumor, antipyretic and immune enhancement effects, known as "traditional Chinese medicine antibiotics", commonly used to treat bacterial infectious diseases [6].
  • Veterinary SHUANGHUANGLIAN ZHUSHEYE is a traditional Chinese veterinary medicine preparation prepared by honeysuckle, skullcap and forsythia three flavors of Chinese herbal medicine through a certain process, its pharmacological effects are mainly antibacterial, antiviral, antipyretic and anti-inflammatory and enhance the body's immunity and other effects, often used to prevent and treat upper respiratory tract infection, acute bronchitis, acute tonsillitis and bacterial infection caused by external wind chill in livestock and poultry, veterinary clinical use is very extensive, the effect is remarkable [7].

References

  1. Wu, Y.H.; Yang, J.Y.; Chen, S.M.; Zhang, X.C.; Chen, Y.W.; Wang, J.Y.; Zhao, S.S.; Lv, B. Preliminary Study on the Effect, Mechanism and Safety of Compound Chinese Medicine in the Treatment of Chicken Colibacillosis. Journal of Shanxi Agricultural Sciences. 2021, 3, 357-360+365.(In Chinese).
  2. Zhou, S.M.; Lu, Y.T.; Zhu, M.; Liang, W.W.; Liu, G.M.; Wei, Y.Y.; Hu, T.J. Curative Effect of Sanhuanglian Preparation on Streptococcosis in Nile Tilapia. Chinese Journal of Fisheries.2021, 3, 49-54. (In Chinese).
  3. Yu, B.; Zhou, J.R.; Jiang, L.L.; Yang, L.; Liu, J.; Zhang, T.; Xu, J.E.; Yu, G.F.; Zou, M.H.; Wu, W.H. Application of antibacterial Chinese veterinary medicine in preventive health care of cattle and sheep breeding. Shanghai Journal of Animal Husbandry and Veterinary Medicine. 2020, 4, 26-27+30.(In Chinese).
  4. Liu, j.; Zhu, S.S.; Zhang, X.F.; Huang, J,P.; Peng, L.; Zhang, X.P.; Cai, Y.Q. An overview of the research on the prevention and treatment of bacterial diseases in chickens by Chinese herbal medicine. Chinese Journal of Traditional Veterinary Science. 2013, 5, 54-57.(In Chinese).
  5. Zhao, Y.; Hu, J.; Zhang, G.M.; Guan, Y.X.; Cheng, G.L.; Qu, H.H. Origin and application of Jingfang Baidu powder. Global Traditional Chinese Medicine. 2020,1, 1996-2002. (In Chinese).
  6. Tao, S.R.; Wang, Y.T.; Liao, C.Y.; Niu, J.L.; Ming, Y.Y.; Ma, Y.; Peng, J.J. Bacteriostatic effects of Siweichuanxinlian powder and its components on Salmonella in vitro. Journal of Traditional Chinese Veterinary Medicine. 2020, 5, 64-67. (In Chinese).
  7. Wang, T.K. The efficacy of several drugs for the treatment of chicken Escherichia colidisease was observed. Chinese Journal of Traditional Veterinary Science. 2019, 4, 78-80. (In Chinese).

Point 3: Rothia was confirmed in livers, lungs and spleens and other organs, but no information is provided regarding which other organs;

Response 3: Thank you very much for your valuable comments. Sorry professor, my description here is inaccurate. I feel sorry for the confusion caused to you. "and other organs" is a habit I developed when I wrote Chinese papers. In Chinese oral communication, for example, today there are only three kinds of dishes in the canteen, A, B, and C, and when people ask me what dishes are in the cafeteria, I will say that the canteen has A, B, C and other dishes. Of course, this kind of expression should not appear in rigorous essay writing. Follow your comments, I have modified my manuscript based on your comments, as shown in Lines 126 and 362.

Point 4: Little information is also provided regarding the methods, for example bacterial genomic extraction kits were used but there is no information on which were used.

Response 4: Thank you very much for your valuable comments. Sorry professor, here is my incomplete description. Follow your comments, I have changed the bacterial DNA extraction kit to the TIANamp Bacterial DNA extraction kit, as shown in Line 159.

Point 5: on section 2.7, the authors refer how animals were monitored and refer the number of deaths as an outcome. This is not acceptable. Were humane endpoints applied?

Response 5: Thank you very much for your valuable comments. Yes professor, I applied the humane endpoint. Laboratory animals have made great sacrifices for human health research, and when they must be infected with various virulent and pathogenic pathogens for scientific research experiments, researchers have the obligation to take the most appropriate and reasonable way possible to reduce the suffering of animals. I originally planned a experiment cycle of 7 days, the mental states of the infected chicks were observed, and appearance changes and clinical symptoms were recorded. Unexpectedly, Five days after infection, the 3×108 CFU chicks in the intraperitoneal injection group died, I immediately carried out the humane endpoints on the experimental group chicks. When the experiment is approaching the end of the experiment, the animal is about to suffer irrelievable or severe pain and discomfort, sometimes death, and the humane endpoints should be used instead of the experimental endpoint. I feel guilty about the death of chicks in the experiment, but I think this result should appear in the manuscript as a result of the experiment. If you feel that this is inappropriate, I will respect and take your comments to make changes to the manuscript. Follow your comments, I realized that it was inappropriate to use the number of deaths as an observational indicator, and I have restated it here, as shown in Lines 240, 241, 243, 244.

Point 6: Chickens were inoculated orally and intraperitoneally? What is the relevance of this second route as this is unlikely to be the cause of the natural infection in farms? Did it provided any useful information for disease control or treatment?

Response 6: Thank you very much for your valuable comments. With the rapid development of biomedical research, the importance of laboratory animals in scientific research has become more and more prominent. Animal experiments are an important process for scientific research, a means that must be used in life science research, and play a very important role in the development of biomedicine. Laboratory animals have made a significant contribution to scientific research. Due to the abundance of peritoneal membrane and large and small omental vessels, the absorption area is large, which is conducive to bacterial infection of experimental animals, so the injected intraperitoneally method is widely used in animal regression tests and mouse pathogenicity tests [1,2]. This study infects laboratory animals using orally and injected intraperitoneally as reference to Kang et al.[1], and citations have been added to the manuscript, as shown in Lines 233-235.

References

  1. Kang, Y.H.; Zhou, H.S.;Jin, W.J. Rothia nasimurium as a Cause of Disease: First Isolation from Farmed Geese. Veterinary Sciences. 2022, 9, 197.
  2. Paul,; Isore,D.P.; Joardar, S.N.; Mukhopadhayay, S.K.; Ganguly, S. Pathogenicity test of Staphylococcus sp. by experimental animal inoculation technique. Indian Journal of Comparative Microbiology, Immunology and Infectious Diseases. 2014, 35, 95-96.

Point 7: Focus on the causes of the outbreak could be higher as possibly low welfare levels were at the origin of the outbreak as this is an opportunistic disease. The impact that poor housing conditions may have on animal welfare was not discussed and possible treatments with antibiotics, that should be avoided, were the main focus. Not only sanitation is important but also animal density and the conditions how animals are housed.

Response 7: Thank you very much for your valuable comments. I am very much agree with your professional point of view. Focus on the causes of the outbreak could be higher as possibly low welfare levels were at the origin of the outbreak. Unfortunately, due to the current situation in our school, we are unable to verify the causes of the outbreak. Follow your comments, I added to the discussion about the impact that poor housing conditions may have on animal welfare, as shown in Lines 421-429. I am very much agree with your another opinion, possible treatments with antibiotics, that should be avoided, were the main focus. Relevant content is discussed in the manuscript, as shown in Lines 438-445.

Point 8: Recommendations for non terminal sampling could be given or to prevent outbreaks of this disease could have been provided.

Response 8: Thank you very much for your valuable comments. Follow your comments, I have added recommendations for preventing outbreaks of this disease, as shown in Lines 429-438.

Point 9: Very little information about the housing conditions at the farm and at the lab were provided.

Response 9: Thank you very much for your valuable comments. This study was conducted at Key Laboratory of Tarim Animal Husbandry Science and Technology, Xinjiang Production & Construction Corps. Follow your comments, I have added relevant information about the housing conditions at the farm, as shown in Lines 254-257.

In addition, I have corrected carefully the several minor mistakes and inaccurately expressed, as shown in Lines 98, 109, 440, 445, 462, 465, 468, 529, 552, 623.

I sincerely thank you and the reviewers for their enthusiastic work, and desire that the correction will meet with your approval. Once again, I would like to thank the referee again for taking the time to review our manuscript.

Thank you and best regards.

Yours sincerely

author:

Jiahao Zhang

Name:lianrui Li

Reviewer 3 Report (New Reviewer)

The manuscript is correctly written. Material and methods and results are presented well. But in my opinion, the manuscript is very similar to the study published in Veterinary Sciences, 2022 by Kang Y. et al. „Rothia nasimurium as cause of disease: first isolation from farmed geese.  Nonetheless, the important part of the study was the identification of resistance genes in the R. nasimurium genome.

Animal experiment and its results: In this experiment blood and liver parameters and immune status should be included in the treatment group.  

Discussion:

A more comprehensive discussion should be included, such as comparing the results of Kang et(2022). 

Other remarks:

89-90„ However, according to the limited reports on the pathogenicity and drug resistance of Rothia nasimurium, it often shows high levels of resistance to a variety of commonly used antibiotics” - lack of citations

99-100: As the such, investigation of the mechanism of Rothia nasimurium drug resistance is of great clinical significance and would provide guidance for the development of new drugs” – way mechanisms of drug resistance are great clinical significance ?. And what new drugs (antibiotics) can be developed based on the drug resistance of the R. nasimurium? It should be discussed.

Figure 4. „Phylogenetic tree based on bacterial 16S rRNA genes” will be improved (quality of figure)

407-408” However, there are few reports on the high-level drug resistance of Rothia nasimurium…..” should be more discussed and lack of citations

Kang et al. [7] et al. found that Rothia nasimurium can cause severe feather loss in goslings…” mistake in citation position (should be 8).

Author Response

Response to Reviewer 3 Comments

Dear Reviewer,

On behalf of my co-authors, I appreciate you very much for your positive and constructive comments and suggestions on our manuscript entitled“Rothia nasimurium as a Cause of Disease: First Isolation from Farmed Chickens” (ID:vetsci-1897783). The comments are valuable for revising the paper, as well as of important guiding significance to our researches. I have studied comments carefully and have made corresponding correction. Revised portions are made in red in the revised manuscript. Once again, I sincerely thank you for your constructive and positive comments.

I have tried our best to revise the manuscript according to your comments. Attached please find the revised version, which I would like to submit for your kind consideration. The main corrections in the paper and the responses to your comments are as follows:

Point 1: Animal experiment and its results: In this experiment blood and liver parameters and immune status should be included in the treatment group.

Response 1: Thank you very much for your valuable comments. Professor you are very right, your proposed method is very good, your rigorous attitude and erudite knowledge make me admire! Unfortunately, we were unable to carry out your proposal due to objective conditions. Due to the epidemic prevention policies of our cities and schools, we are unable to leave the school and nothing from the outside can come in. I apologize for this and ask for your forgiveness and understanding. Thank you again for your valuable comments.

Point 2: A more comprehensive discussion should be included, such as comparing the results of Kang et(2022).

Response 2: Thank you very much for your valuable comments. Follow your comments, I have added content accordingly to the discussion, as shown in Lines 392-395.

Point 3: 89-90: However, according to the limited reports on the pathogenicity and drug resistance of Rothia nasimurium, it often shows high levels of resistance to a variety of commonly used antibiotics” - lack of citations.

Response 3: Thank you very much for your valuable comments. Thank you so much for your careful check. I am very sorry for my negligence. Follow your comments, I have already added citations after this sentence, as shown in Line 92.

Point 4: 99-100: As the such, investigation of the mechanism of Rothia nasimurium drug resistance is of great clinical significance and would provide guidance for the development of new drugs” – way mechanisms of drug resistance are great clinical significance ?And what new drugs (antibiotics) can be developed based on the drug resistance of the R. nasimurium? It should be discussed.

Response 4: Thank you very much for your valuable comments. Sorry professor, I did not translate this sentence accurately and did not express my meaning accurately. I feel sorry for the confusion caused to you. What I am trying to say here is that it is particularly important to monitor the drug resistance and drug resistance mechanism of Rothia nasimurium. Regarding the question of which new drugs (antibiotics) can be developed based on the resistance of the Rothia nasimurium, due to my limited knowledge and the shallow level of this study, I cannot give you a satisfactory answer at present. Please forgive me. I will continue to study hard and do more in-depth research in the future. I've reworded it here, as shown in Lines 100-104.

Point 5: Figure 4: Phylogenetic tree based on bacterial 16S rRNA genes” will be improved (quality of figure)

Response 5: Thank you very much for your valuable comments. Follow your comments, I have replaced Figure 4 to enhance the quality of the picture, as shown in Figure 4. In addition, I have sent Figure 4 as an attachment to the Editorial office.

Point 6:407-408” However, there are few reports on the high-level drug resistance of Rothia nasimurium…..” should be more discussed and lack of citations.

Response 6: Thank you very much for your valuable comments. I have revised my manuscript based on your comments to add discussed and citations to this section, as shown in line 420.

Point 7: Kang et al. [7] et al. found that Rothia nasimurium can cause severe feather loss in goslings…” mistake in citation position (should be 8).

Response 7: Thank you very much for your valuable comments. Thank you so much for your careful check. I am very sorry for my negligence. I've modified the citation position of Kang et al.[8], as shown in Line 489.

In addition, I have corrected carefully the several minor mistakes and inaccurately expressed, as shown in Lines 98, 109, 440, 445, 462, 465, 468, 529, 552, 623.

I sincerely thank you and the reviewers for their enthusiastic work, and desire that the correction will meet with your approval. Once again, I would like to thank the referee again for taking the time to review our manuscript.

Thank you and best regards.

Yours sincerely

author:

Jiahao Zhang

Name:lianrui Li

Round 2

Reviewer 1 Report (Previous Reviewer 2)

Dear Authors,

The manuscript may be accepted in present form.

Author Response

Response to Reviewer 1 Comments

Dear Reviewer,

On behalf of my co-authors, I appreciate you very much for your positive and constructive comments and suggestions on our manuscript entitled“Rothia nasimurium as a Cause of Disease: First Isolation from Farmed Chickens” (ID:vetsci-1897783). The comments are valuable for revising the paper, as well as of important guiding significance to our researches. Once again, I sincerely thank you for your constructive and positive comments.

I would like to express my gratitude to the professor for his recognition and acceptance of this manuscript.

Thank you and best regards.

Yours sincerely

author:

Jiahao Zhang

Name:lianrui Li

Reviewer 2 Report (New Reviewer)

All comments were addressed. For the chinese medicine use, it would be useful to state on the manuscript that "chinese traditional medicine was used, based on natural ingredients such as...".

Author Response

Response to Reviewer 2 Comments

Dear Reviewer,

On behalf of my co-authors, I appreciate you very much for your positive and constructive comments and suggestions on our manuscript entitled“Rothia nasimurium as a Cause of Disease: First Isolation from Farmed Chickens” (ID:vetsci-1897783). The comments are valuable for revising the paper, as well as of important guiding significance to our researches. I have studied comments carefully and have made corresponding correction. Revised portions are made in red in the revised manuscript. Once again, I sincerely thank you for your constructive and positive comments.

I have tried our best to revise the manuscript according to your comments. Attached please find the revised version, which I would like to submit for your kind consideration. The main corrections in the paper and the responses to your comments are as follows:

Point 1: All comments were addressed. For the chinese medicine use, it would be useful to state on the manuscript that "chinese traditional medicine was used, based on natural ingredients such as...".

Response 1: Thank you very much for your valuable comments. Follow your comments, I have modified my manuscript, as shown in Lines 60-65. The following is added : Jingfang Baidu San is based on natural ingredients such as Schizonepeta Tenuifolia Brip, Saposhnikoviae Radix, and Bupleurum, etc. Siwei Chuanxinlian San is based on natural ingredients such as Andrographis Paniculata, Isatidis Folium, Polygonum hydropiper, and Tadehagi Triquetrum. SHUANGHUANGLIAN ZHUSHEYE is based on natural ingredients such as Lonicera Japonica Thunb, Scutellaria Baicalensis, and Forsythia Suspensa. Thank you again for your valuable comments.I sincerely hope that my correction will meet with your approval.

I sincerely thank your enthusiastic work, and desire that the correction will meet with your approval. Once again, I would like to thank you again for taking valuable time to review our manuscript.

Thank you and best regards.

Yours sincerely

author:

Jiahao Zhang

Name:lianrui Li

Reviewer 3 Report (New Reviewer)

Accept in present form. All suggestions have been included in manuscript.

Author Response

Response to Reviewer 3 Comments

Dear Reviewer,

On behalf of my co-authors, I appreciate you very much for your positive and constructive comments and suggestions on our manuscript entitled“Rothia nasimurium as a Cause of Disease: First Isolation from Farmed Chickens” (ID:vetsci-1897783). The comments are valuable for revising the paper, as well as of important guiding significance to our researches. Once again, I sincerely thank you for your constructive and positive comments.

I would like to express my gratitude to the professor for his recognition and acceptance of this manuscript.

Thank you and best regards.

Yours sincerely

author:

Jiahao Zhang

Name:lianrui Li

This manuscript is a resubmission of an earlier submission. The following is a list of the peer review reports and author responses from that submission.

Round 1

Reviewer 1 Report

You are correct much more work is required to demonstrate this is  even a potential pathogen.  I would continue the great work, but we need more evidence I am afraid.   This would make an interesting short communication of concern.  But this is not a major health risk at the moment and certainly not to man.  There is too much alarmist language in your paper.

Author Response

Response to the reviewers' comments

Dear Reviewer,

        On behalf of my co-authors, I appreciate you very much for your positive and constructive comments and suggestions on our manuscript entitled“Rothia nasimurium as a Cause of Disease: First Isolation from Farmed Chickens” (ID:vetsci-1897783). The comments are valuable for revising the paper, as well as of important guiding significance to our researches. I have studied comments carefully and have made corresponding correction. Revised portions are made in red in the revised manuscript. Once again, I sincerely thank you for your constructive and positive comments.

        I have tried our best to revise the manuscript according to the comments. Attached please find the revised version, which I would like to submit for your kind consideration. The main corrections in the paper and the responses to your comments are as follows:

Point 1: Medicine?

Response 1: Thank you very much for your valuable comments. Yes, I mean Medicine. I have changed drug to antibiotic, as shown in Lines 14, 77, 302, 320, 384, 417, 466-467.

Point 2: Given that the organism is sensitive to pencillin why was this not used?

This is one of the most common antibitoics and known to be highly effective against Micrococcus?

Response 2: Thank you very much for your valuable comments. As you commented, this is quite true, penicillin is one of the most common antibiotics and known to be highly effective against Micrococcus. For the first question,This section describes how farmers and veterinarians treat sick chickens before we diagnose them. At that time, no susceptibility tests were done, and they did not know that penicillin was sensitive to this organism. For the second question, professor you are correct , penicillin is one of the most common antibiotics and has a strong effect on gram-positive bacteria. But you know, the medication regimen in an area is often determined by the pathogen of the disease and the local veterinarian. After consulting information and asking local veterinarians, 20 years ago, chickens in this area had many gram-positive infections represented by staphylococcus, and at that time, penicillin was a common antibiotic for the local treatment of chicken diseases. In recent years, there have been few reports of breeding chickens infected with Gram-positive bacteria in the region. In addition, for mitigate bacterial resistance, Chinese medicine ingredient drugs are now preferred for the treatment of sick chickens in this region, and antibiotics are considered if the treatment is ineffective. Therefore, after the onset of chicken disease, penicillin was not used for treatment at the first time. I've reworded it here, as shown in Lines 30-31.

Point 3: 0.5-2.5% mortality is not high mortaltiy?

Response 3: Thank you very much for your valuable comments. Yes,0.5-2.5% mortality is not high mortaltiy. Here's what I'm saying is inaccurate. I have deleted "and many died", as shown in Line 54.

Point 4: The vet treats?? the farmer should have asked?

Response 4: Thank you very much for your valuable comments. In recent years, the scale of chicken farming in the region has expanded year by year, and it has become an important poverty alleviation industry in the region. However, most free-range farmers have a low level of Feeding management and the feeding environment is poor. When the Farmed Chickens is sick, usually in order to save money, farmers will habitually use several of their commonly used drugs to treat the sick chicken, if the treatment is not effective will find a veterinarian to diagnose. I've reworded it here, as shown in Lines 58-62.

Point 5: Gentamcin not effective against systemci diseases?

Response 5: Thank you very much for your valuable comments. According to inquiries from farmers and veterinarians, in this case, there was no significant improvement in the condition of the sick chickens after they were treated with Gentamicin. In addition, the results of the susceptibility test of this study showed that Gentamicin showed severe resistance to Rothia nasimurium. Studies Kang et al.[1] and Wang et al.[2] also yielded the same results. I've reworded it here, as shown in Lines 62-64.

  • Kang, Y.H.; Zhou, H.S.;Jin, W.J. Rothia nasimurium as a Cause of Disease: First Isolation fromFarmed Geese. Veterinary Sciences. 2022, 9, 5
  • Wang, M.; Li, Y.; Lin, X.; Xu, H.; Li, Y.; Xue, R.; Wang, G.; Sun, S.; Li, J.; Lan, Z.; et al. Genetic characterization, mechanisms and dissemination risk of antibiotic resistance of multidrug-resistant Rothia nasimurium. Infect. Genet. Evol. 2021, 90, 104770.

Point 6: substantial economic losses.

Response 6: Thank you very much for your valuable comments. Thank you so much for your careful check. My description here is inaccurate. I've reworded it here, as shown in Lines 31, 63.

Point 7: Computer confirmation, not convienced.

Response 7: Thank you very much for your valuable comments. My description here is inaccurate. I've reworded it here, as shown in Line 67.

Point 8: Need more evidence this is a commensal Scaremongering. NO massive extrapolation.

Response 8: Thank you very much for your valuable comments. This sentence is quoted from the paper of Wang et al. [1]. You're correct, Need more evidence this is a commensal Scaremongering. I've reworded it here and some of the content was deleted, as shown in Lines 86, 92.

  • Wang, M.; Li, Y.; Lin, X.; Xu, H.; Li, Y.; Xue, R.; Wang, G.; Sun, S.; Li, J.; Lan, Z.; et al. Genetic characterization, mechanisms and dissemination risk of antibiotic resistance of multidrug-resistant Rothia nasimurium. Infect. Genet. Evol. 2021, 90, 104770.

Point 9: Seriously? Not enough

Response 9: Thank you very much for your valuable comments. Thank you so much for your careful check. Here I am not precise in my wording. I've reworded it here, as shown in Lines 103, 453 and 470.

Point 10: No they are not in animal production.

Response 10: Thank you very much for your valuable comments. I am very sorry for my negligence. Here is my mistake. I've reworded it here, as shown in Lines 182 and 319.

Point 11: So just a typical poor chinese poultry far.

Response 11: Thank you very much for your valuable comments. professor, you are very correct. Here I have made too many statements in order to explain the situation of the poultry farm. I've reworded it here, as shown in Lines 249-253.

Point 12: Odd bacterial plate. 

Response 12: Thank you very much for your valuable comments. Follow your comments, I've replaced another bacterial plate, as shown in Figure 1.

Point 13: You are correct much more work is required to demonstrate this is even a potential pathogen. I would continue the great work, but we need more evidence I am afraid. This would make an interesting short communication of concern. But this is not a major health risk at the moment and certainly not to man. There is too much alarmist language in your paper.

Response 13: Thank you very much for your valuable comments. professor you are correct, this is not a major health risk at the moment and certainly not to man. What I'm trying to say is that it's a potential pathogen and We should pay attention to its potential harm. Here I use inaccurate words, there are too many alarmist language. I feel sorry for the inconvenience brought to you. I've reworded it here, as shown in Lines 389-410.

        In addition, I have corrected carefully the several minor mistakes and inaccurately expressed, as shown revised portions marked in red. 

        I sincerely thank you for your enthusiastic work, and hope that the correction will meet with your approval. Special thanks to you for your good comments.

Thank you and best regards.

Yours sincerely

author:

Jiahao Zhang

Name:lianrui Li

Reviewer 2 Report

Dear Authors,

This is an interesting paper, with an important finding of recognizing a rare aetiological agent in diseased chickens. However, some clarifications are needed.

Line 114: My major concern: You describe that you repeated culturing process until a purified colony was obtained. This means that you had a mixed culture at the start? If so, you have to describe: Which are the other bacteria cultured from the sample? Did you identify them? How did you conclude that these other bacteria were not the aitiological agent of the disease?

Line 170: Describe phylogenetic analysis methodology.

Some serious editing in English must take place:

Examples:

Line 14: change bacteria to bacterium and this must be changed in the whole manuscript. Bacteria is the plural to bacterium.

Line 20: identified to be

Line 27: The genus is Rothia, the family Micrococcaceae. Same to Line 64

Line 105: Change unwell to clinically ill

Line 132: You probably mean after staining, no after natural drying

Line 139: Rephrase, this sentence is incorrect. e.g. The strain was cultured for 72 hour to perform biochemical tests.

Line 209: delete the ' before MDV and correct AILT to ILTV

Line 239: The pathogen name always in italics.

Line 282: Change drug-susceptible tablet to antibiotic diffusion disk

Line 312: Change to "Left: hair condition of control group..." and refer that this is 3 days after the infection.

Line 315: Refer that Figure 5C is 4 days after the infection.

Line 344: Animal flora? What do you mean? Other bacteria of the animal microbiome? Rephrase

Line 353: Acinetobacter baumannii to italics here and eveywhere else

Author Response

Response to the reviewers' comments

Dear Reviewer,

         On behalf of my co-authors, I appreciate you very much for your positive and constructive comments and suggestions on our manuscript entitled“Rothia nasimurium as a Cause of Disease: First Isolation from Farmed Chickens” (ID:vetsci-1897783). The comments are valuable for revising the paper, as well as of important guiding significance to our researches. I have studied comments carefully and have made corresponding correction. Revised portions are made in red in the revised manuscript. Once again, I sincerely thank you for your constructive and positive comments.

        I have tried our best to revise the manuscript according to the comments. Attached please find the revised version, which I would like to submit for your kind consideration. The main corrections in the paper and the responses to your comments are as follows:

Point 1: Line 114: My major concern: You describe that you repeated culturing process until a purified colony was obtained. This means that you had a mixed culture at the start? If so, you have to describe: Which are the other bacteria cultured from the sample? Did you identify them? How did you conclude that these other bacteria were not the aitiological agent of the disease?

Response 1: Thank you very much for your valuable comments. Sorry professor, my description here is inaccurate. I did not grow other bacteria from the sample. I feel sorry for the confusion causedto to you. I've reworded it here, as shown in Lines 119-122.

Point 2: Line 170: Describe phylogenetic analysis methodology.

Response 2: Thank you very much for your valuable comments. Follow your comments, I have added describe phylogenetic analysis methodology in corresponding locations, as shown in Lines 175-180.

Point 3: Line 14: change bacteria to bacterium and this must be changed in the whole manuscript. Bacteria is the plural to bacterium.

Response 3: Thank you very much for your valuable comments. Follow your comments, I have modified my manuscript based on your comments, as shown in Lines 15, 24, 39, 41, 47, 90, 94, 96, 105, 160, 192, 255, 262, 267, 274, 279, 289-291, 293, 294, 354, 355, 357, 359, 379, 412, 415, 465, 470, 471, .

Point 4: Line 20: identified to be

Response 4: Thank you very much for your valuable comments. Follow your comments, I have changed identified to identified to be, as shown in Line 19.

Point 5: Line 27: The genus is Rothia, the family Micrococcaceae. Same to Linse 26-27 and 69 .

Response 5: Thank you very much for your valuable comments. I am very sorry for my negligence. Follow your comments, I have changed Rothella to Rothia, as shown in Lines 26-27 and 70.

Point 6: Line 105: Change unwell to clinically ill.

Response 6: Thank you very much for your valuable comments. Follow your comments, I have changed unwell to clinically ill, as shown in Line 110.

Point 7: Line 132: You probably mean after staining, no after natural drying.

Response 7: Thank you very much for your valuable comments. I am very sorry for my negligence, my description here is inaccurate. I've reworded it here, as shown in Lines 141-143.

Point 8: Line 139: Rephrase, this sentence is incorrect. e.g. The strain was cultured for 72 hour to perform biochemical tests.

Response 8: Thank you very much for your valuable comments. I rewrote this section as you suggested, as shown in Line 148.

Point 9: Line 209: delete the ' before MDV and correct AILT to ILTV.

Response 9: Thank you very much for your valuable comments. Follow your comments, I have deleted the ' before MDV and changed AILT to ILTV, as shown in Line 247. Also, considering that Mycoplasma synoviae (MS) is not a virus, therefore, I removed MS in the Virus Detection and Corresponding Results section.

Point 10: Line 239: The pathogen name always in italics.

Response 10: Thank you very much for your valuable comments. Thank you so much for your careful check. I have modified my manuscript based on your comments, as shown in Lines 274-275.

Point 11: Line 282: Change drug-susceptible tablet to antibiotic diffusion disk.

Response 11: Thank you very much for your valuable comments. Follow your comments, I have changed drug-susceptible tablet to antibiotic diffusion disk, as shown in Lines 323-324.

Point 12: Line 312: Change to "Left: hair condition of control group..." and refer that this is 3 days after the infection.

Response 12: Thank you very much for your valuable comments. Follow your comments, I have changed control group chicks to hair condition of control group chicks, and refer that this is 3 days after the infection, as shown in Lines 365-366.

Point 13: Line 315: Refer that Figure 5C is 4 days after the infection.

Response 13: Thank you very much for your valuable comments. Follow your comments, I have stated Figure 5C and 5D is 4 days after infection and Figure 5E is 5 days after infection, as shown in Lines 368-371.

Point 14: Line 344: Animal flora? What do you mean? Other bacteria of the animal microbiome? Rephrase

Response 14: Thank you very much for your valuable comments. I am very sorry for my miswriting, my description here is inaccurate. I feel sorry for the confusion causedto to you. I've reworded it here, as shown in Lines 395-397.

Point 15:Line 353: Acinetobacter baumannii to italics here and eveywhere else.

Response 15: Thank you very much for your valuable comments. Thank you so much for your careful check. I have modified my manuscript based on your comments, as shown in lines 405-406, 567, 574.

         In addition, I have corrected carefully the several minor mistakes and inaccurately expressed, as shown revised portions marked in red. 

         I sincerely thank you for your enthusiastic work, and hope that the correction will meet with your approval. Special thanks to you for your good comments.

Thank you and best regards.

Yours sincerely

author:

Jiahao Zhang

Name:lianrui Li
